# Thermal and Mechanical Characterization of Epoxy/Polyimide Blends via Postcuring Process

**DOI:** 10.3390/polym15051072

**Published:** 2023-02-21

**Authors:** Yong-Min Lee, Kwan-Woo Kim, Byung-Joo Kim

**Affiliations:** 1Convergence Research Division, Korea Carbon Industry Promotion Agency, Jeonju 54852, Republic of Korea; 2Department of Carbon-Nanomaterials Engineering, Jeonju University, Jeonju 55069, Republic of Korea

**Keywords:** polymer blends, epoxy, polyimide, thermal properties, mechanical properties

## Abstract

In this study, the effects of polyimide (PI) content and postcuring on thermal and mechanical properties in PI and epoxy (EP) blending systems were investigated. EP/PI (EPI) blending reduced the crosslinking density and improved the flexural and impact strength due to ductility. On the other hand, in the postcuring of EPI, the thermal resistance improved due to the increased crosslinking density and the flexural strength increased by up to 57.89% due to the enhanced stiffness, but the impact strength decreased by up to 59.54%. EPI blending induced the improvement in the mechanical properties of EP, and the postcuring process of EPI was shown to be an effective method to improve heat resistance. It was confirmed that EPI blending induces improvement in the mechanical properties of EP, and the postcuring process of EPI is an effective method for improving heat resistance.

## 1. Introduction

Polymer matrix composites are increasingly used in the military, aviation, and sports leisure industries due to their lightweight nature as compared to metals and ceramics [1,2]. Polymers are primarily classified into thermosetting and thermoplastic ones. The use of recyclable thermoplastics in recent years has significantly increased due to environmental and sustainability issues. However, applications of thermoplastic polymers are limited because they have lower mechanical properties than thermosetting polymers. Additionally, a thermosetting resin generally has high mechanical strength because the linear structure can form a network through curing reactions in the presence of a curing agent [1,2,3].

Among the various thermosetting resins (such as epoxy, phenolic formaldehyde, and unsaturated polyester resins), epoxy resins are most commonly used due to their excellent heat resistance, corrosion resistance, high adhesion strength, and electrical insulation properties. The bisphenol A-based epoxy consisting of two epoxy groups is a representative of epoxy resins [3,4]. However, its brittleness is a significant disadvantage in that it can be easily destroyed by impact forces [5,6]. To compensate for this shortcoming, a polymer blending technology that mixes two or more polymers has been previously investigated [7,8]. Conventionally, to improve the brittleness of an epoxy resin, a method of adding an elastomer such as carboxyl-terminated butadiene-acrylonitrile (CTBN) [9,10] or amine-terminated butadiene-acrylonitrile (ATBN) [11] has been studied. It has been reported that adding an elastic body can rapidly decrease heat resistance and mechanical strength, although the brittleness of the epoxy resin is improved [9,10,11]. Therefore, flexible polymers with high mechanical strength such as the thermoplastic engineering polymers polysulfone [12], polyester [13], polyamide [4,14,15], and even poly(ether imide) [16,17,18] have been mixed with epoxy resin. These blends have also been reported to show improved heat resistance and mechanical strength than existing elastomer blends. However, thermosetting/thermoplastic blends have some limitations in inducing high-strength mechanical properties because the mechanical strength of the blend is determined only by the structure of the physical entanglement between the polymer chains. Therefore, previous studies have attempted to improve the brittleness of conventional epoxy by blending it with various thermosetting polymers. In general, thermosetting/thermosetting blends have high crosslinking densities during curing. Thus, they have excellent heat resistance and mechanical strength. However, their brittleness remains a problem to be solved [19]. Recently, research on synthesizing and using epoxidized poly(2-eugenoxyethyl vinyl ether) as an alternative to DGEBA EP has been conducted. Unlike the existing highly brittle EP, the modulus of elasticity can be adjusted, but the synthesis process is complicated [20].

Polyimide (PI) is a well-recognized engineering thermosetting polymer that has two forms depending on its molecular structure: a linear one and a heterocyclic one. A double aromatic heterocyclic polyimide is a commercial product with excellent mechanical, thermal, and insulating properties [21,22,23,24]. PI has reactive groups in its molecular chain that can chemically react (or cure) with epoxy resin. It also has a molecular structure that is more flexible than epoxy resin. Therefore, a blend of epoxy/polyimide can lower the elastic modulus and lead to high mechanical strength. However, the thermal and mechanical properties of epoxy/polyimide blends have not been thoroughly or exhaustedly reported yet. Therefore, this study aimed to investigate the effects of varying the polyimide and epoxy resin ratio and postcuring on the thermal and mechanical properties of epoxy/polyimide blends.

## 2. Materials and Methods

### 2.1. Materials

A bifunctional epoxy resin (EP, diglycidyl ether of bisphenol A, Kukdo Chemical, Seoul, Republic of Korea, YD-128, EEW of 184–190 g/eq, viscosity of 11,500–13,500 cps at 25 °C) was used as the main matrix in this study. Polyimide (PI, Dongbaek Fine-Chemical, Busan, Republic of Korea, DFPI-101, T_g_ of 320 °C, flexural strength of 180 MPa, AHEW of 294.30 g/eq) was blended at different ratios. The polyimide used in this work was mixed with N-methyl-2-pyrrolidone (NMP, Sigma-Aldrich, Saint Louis, MO, USA) at a ratio of 82/18 wt.%. An amine-curing agent (DDM, 4,4′-diaminodiphenyl methane, Sigma-Aldrich, Saint Louis, MO, USA, AHEW of 49.565 g/eq, T_m_ of 89 °C) was used as the EP curing agent. The chemical structures of EP, PI, and DDM are shown in Appendix A. In addition, the TGA thermograms of PI and NMP are shown in Appendix A.

### 2.2. Sample Preparation

EP resin as the main matrix was blended with PI in the range of 0 to 50 phr. The nomenclature of the prepared samples is shown in Appendix A. EP and PI were added into a beaker of 500 mL which was then placed in a heating mantle at 70 °C. After stirring at 700 rpm using a mechanical stirrer (the centrifugal type with a Teflon blade) for 30 min, DDM, a powdered curing agent, was added. The mixture was then stirred again at 700 rpm for 30 min at 70 °C. The amount of DDM added was calculated based on the stoichiometric equivalent ratio as follows:(1)$$EEW=\frac{M_{w}}{N_{f}}\;\mathrm{(g/eq)}$$
(2)$$AHEW=\frac{M_{w}}{N_{h}}\;\mathrm{(g/eq)}$$
(3)$$Curing\;agent\;amount=\frac{\left(E_{w}\times AHEW\right)}{EEW}\;\mathrm{(g)}$$
where *EEW* (g/eq), *M_w_* (g/mol), *N_f_*, *N_h_*, and *E_w_* (g) denote the EP equivalents, molecular weights of each substance, epoxide number, active hydrogen number, and EP addition amount in the above equations, respectively.

After completing the stirring process, the mixture was defoamed at 70 °C for 20 min using a vacuum oven. The mixture was cured at 170 °C for one hour at a heating rate of 5 °C/min using a convection curing oven. In addition, the prepared EPI-5 specimens were postcured using a curing oven at 170 °C (5 °C/min) for 1–10 h. Appendix A illustrates the curing mechanisms of an EP/DDM.

### 2.3. Thermal Analysis of Epoxy/Polyimide

The curing temperature of the EPI mixture was measured using a differential scanning calorimeter (DSC, DSC-60, Shimadzu, Kyoto, Japan). A total of 13 mg of the mixture was placed into an aluminum pan and heated to 30–300 °C (5 °C/min) in a nitrogen atmosphere at a feeding rate of 50 cc/min. In addition, the glass transition temperature (T_g_) and curing degree of the postcured EPI was analyzed by placing a specimen of about 4.37 mg in an aluminum pan and heated to 30–300 °C (5 °C/min) in a nitrogen atmosphere at a feeding rate of 50 cc/min.

Thermogravimetric analysis (TGA, TGA-50, Shimadzu, Kyoto, Japan) can measure the initiating decomposition temperature, decomposition propagation behaviors, and total thermal stability in various atmospheres such as nitrogen, helium, and argon. The heat resistance of the EPI was measured by TGA. Thirteen milligrams of crushed blend samples (after curing) were placed into an alumina pan and heated to 900 °C (5 °C/min) in a nitrogen atmosphere at a feeding rate of 50 cc/min. The initial pyrolysis temperature (IDT), maximum weight loss temperature (T_max_), thermal stability index (A*·K*), and integral pyrolysis temperature (IPDT) were calculated based on the results of the TGA.

### 2.4. Thermomechanical Properties of EPI

Polymers are materials that exhibit viscoelastic rheological properties different from those of inorganic materials. Thus, it is necessary to analyze the dynamic mechanical analyzer (DMA, Q800, Waters, Milford, MA, USA) under dynamic stress or deformation conditions [25]. The viscoelastic properties and crosslinking densities of EPI were determined using a DMA in a nitrogen atmosphere with a feeding rate of 50 cc/min at a temperature range of 30 to 300 °C with a heating rate of 5 °C/min and 1 Hz of vibration. Each specimen was manufactured by precise machining into a rectangular parallelepiped with a length of 35 mm, a width of 10 mm, and a height of 2 mm. The crosslinking density of each blend was calculated using Flory’s rubber elasticity theory [26] based on the measured DMA results.

When a polymer is exposed to a high temperature for a long time, its physical properties are directly affected. Thus, a thermomechanical analyzer (TMA, Q400, Waters, Milford, MA, USA) was used to measure its dimension change in terms of the thermal expansion coefficient. The thermal expansion coefficient of each EPI was determined using a TMA in a nitrogen atmosphere with a feeding rate of 50 cc/min for each sample at a heating rate of 5 °C/min in the temperature range of 30 to 300 °C. Specimens were also manufactured by precision processing into a rectangular parallelepiped having a length of 5 mm, a width of 5 mm, and a height of 8 mm.

### 2.5. Mechanical Properties of EPI

For the flexural strength test, specimens of all EPI were manufactured by precisely processing them in a rectangular parallelepiped form with a length of 60 mm, a width of 25 mm, and a thickness of 3 mm according to ASTM D790 [27]. The flexural behaviors (or strengths) were measured using a universal material tester (UTM, Lloyd Instruments, Bognor Regis, UK) with a three-point bending test method. The span was fixed at 48 mm and the crosshead speed was set at 1 mm/min.

For the impact strength test, specimens were also manufactured by precisely processing them in a length of 63 mm, a width of 13 mm, a thickness of 3 mm, and a notch depth of 4 mm according to ASTM D256 [28]. Five impact strength measurements were performed for each sample with an Izod impact strength tester (RESIL Impactor, Instron, Norwood, MA, USA) using a 1.0 J hammer.

After performing a mechanical property analysis of EPI, scanning electron microscopy (SEM, AIS2100C, Seron Technology, Uiwang, Republic of Korea) was used to observe the stress propagation shape of the fracture surface of each specimen. Sputtering was performed with platinum for 3 min to prevent the charging phenomenon. All images were obtained at 1.0 × 10^−5^ torr and 25 kV voltage.

## 3. Results and Discussion

### 3.1. DSC Results of EPI

The DSC curing peak and curing behavior of the EPI mixed with various contents of PI are shown in Figure 1 and Appendix A, respectively, and the DSC analysis result of the cured EPI is shown in Figure 2 and Table 1. Each sample exhibited T_cp_ in the temperature range of 162.39–168.05 °C, and the curing reaction of EP was started (T_ci_) at 126.20 °C, but as PI was added, the curing started at a relatively low temperature compared to EP. In addition, it was confirmed that the temperature range of the exothermic peak was broader than that of pure EP. This was considered to be the result of the epoxide groups of EP reacting first with the hydroxyl groups of PI at a low temperature and then further reacting with the amine groups of DDM. The schematic diagram of the crosslinking density of EP/DDM and EPI above is shown in Appendix A. The crosslinked network of oligomer EP and low molecular weight DDM is brittleness [29,30] due to it is narrow and dense free volume, whereas the crosslinked network of EP and high molecular weight PI is flexible [31] because it is a wide free volume. It was judged that the cured EP, EPI-5, and postcured EPI specimens were all completely cured, and no exothermic peak was observed. The EPI-5 specimen showed an endothermic peak at about 187.47 °C, which was judged to be that the free volume was broadened due to the combination of high molecular weight PI and EP, and the softening temperature appeared due to plasticization. In addition, the T_g_ of EP was observed to be about 160.71 °C, and it was confirmed that the T_g_ was lowered to a maximum of 55.67 °C (EPI-5) due to the blend with PI. On the other hand, as the postcuring time increased, the T_g_ of the EPI-5 specimen increased up to 74.52 °C (EPI-5-10). This suggests that the energy required to move the crosslinked network increased due to the increase in the crosslinking density.

### 3.2. Crosslinking Density of EPI

The storage modulus (tan δ), glass transition temperature (T_g_), and crosslinking density of EPI and postcured EPI were obtained through DMA. The results are illustrated in Figure 3 and Table 2. The storage modulus of the EPI decreased with the increase in temperature. When the value of tan δ of a sample reaches the maximum, the phase delay is maximized, indicating that the phase of the sample changes. This value is recognized as T_g_. The elastic modulus of the polymer in the rubbery plateau region (a region where the elastic modulus is hardly changed above T_g_) is affected by the crosslinking density of the sample [32]. From the DMA results, the crosslinking density was calculated based on Flory’s theory of rubber elasticity [26] as follows:(4)$$v_{e}=\frac{E_{high}^{\prime }}{3RT_{high}}\;(\mathrm{mol/}m^{3})$$
where *E′_high_* (MPa) is the storage modulus in the rubber plateau region, *R* is the ideal gas constant (8.314 J/mol·K), and *T_high_* (K) is the temperature of *E^’^_high_*.

The crosslinking density and T_g_ of the EPI showed a trend to decrease with increasing PI content in the blend, which was lower than that of pure EP. It seems that the distance between the crosslinked network was broadened due to the combination with the PI of long molecular chains, which was judged to decrease the crosslinking density and T_g_. On the other hand, the crosslinking density and T_g_ of the postcured EPI tended to increase, and the crosslinking density was calculated to be similar to or higher than that of the pure EP. It was judged that the postcuring process removed the residue NMP inside the EPI, and the unreacted epoxy reacted to the dense crosslinked structure.

### 3.3. Thermal Expansion Coefficients of EPI

TMA diagrams of EPI are illustrated in Figure 4. The glass transition temperature (T_g_) and the coefficient of thermal expansion (CTE) of each blend were calculated using Equation (5). The results are listed in Table 3.
(5)$$\alpha =\frac{\Delta L}{L_{0}\Delta T}\;\left(\mu m/\mu m{}^{\circ}C\right)$$

Here, Δ*L* (m) is the amount of change in the length, *L*_0_ (m) is the length of the initial specimen, and Δ*T* (°C) is the amount of temperature change.

The T_g_ measured by TMA decreased with increasing PI content in EPI. This seems to be because the crosslinking density of EPI decreased as the PI content increased. However, as the postcuring process progressed, the T_g_ of EPI was shown to increase. The CTE of EPI before T_g_ (glassy region) showed similar values (in the range of 65.20–72.29 × 10^−6^ μm/μm·°C). On the other hand, the CTE value of EPI after T_g_ (rubbery region) changed significantly in proportion to the PI content (169.22 − (−44.57) × 10^−6^ μm/μm·°C). The temperature with a significant dimension change was named the softening temperature (T_s_). As mentioned above, it was judged that the free volume was broadened due to the combination of high molecular weight PI and EP, and the T_s_ appeared due to plasticization. The pure EP showed no T_s_ but decreased from 224.52–186.30 °C with increasing PI content. The CTE values of the postcured EPI before and after T_g_ decreased with an increasing postcuring time (98.44–74.44 and 188.30–174.40 × 10^−6^ μm/μm·°C). The postcuring process induced an increase in T_g_ and T_s_. This increased the crosslinking density of the EPI to improve the elastic modulus and suppressed plasticization, as shown in the gradual dimension change.

### 3.4. Thermal Stability of EPI

The IDT, T_max_, A*·K*, and IPDT of EPI were calculated using Equations (6)–(8). The results are shown in Table 4. IDT refers to the temperature at which the weight decreases more than 5% for the first time. IPDT refers to the total heat required from the initiation to the termination of the decomposition. For A*·K* and IPDT, quantitative values obtained as the area ratio during a single and a multistep decomposition from Doyle [24] were used [14]:(6)$$IPDT=A^{*}\cdot K^{*}\left(T_{f}-T_{i}\right)+T_{i}\;\mathrm{({}^{\circ}C)}$$
(7)$$A^{*}=\frac{S_{1}+S_{2}}{S_{1}+S_{2}+S_{3}}$$
(8)$$K^{*}=\frac{S_{1}+S_{2}}{S_{1}}$$
where *A** is the ratio of the down area of the curve to the total area of the TGA diagram, *K** is the coefficient of *A**, *T_i_* (°C) is the initial temperature, and *T_f_* (°C) is the final temperature.

Each area in the TGA diagram is illustrated in Appendix A. *A** can be expressed as the ratio of the total area of the TGA diagram and the total area of the graph. *K** (a coefficient of *A**) can be expressed as the ratio of the total down area of the curve and the subtracted value (the total down area minus the yield area). *A*·K** is the thermal stability index, with a larger value indicating higher thermal stability [4].

Figure 5a,c illustrates the TGA and DTG analysis results of EPI. EP was rapidly pyrolyzed at about 350 °C. The EPI samples and PI (Appendix A) all had a first mass decrease at about 150 °C. The EPI samples then showed a second mass decrease at about 350 °C while PI showed such a decrease at about 550 °C. The primary change in EPI and PI might have been due to the thermal decomposition of the NMP residue added with PI, unreacted epoxy, and low heat resistance of the blend due to the loose crosslinked structure. In other words, the relatively long molecular chain of the PI might have induced a decrease in the crosslinking density of the EPI. Secondary changes (a drop in mass yield) in all samples might have been due to the decomposition of the main chains of EPI and PI. It is believed that the heat resistance of EPI exhibited a similar level to the neat epoxy resin due to the lack of compactness of the internal molecular structure caused by the low crosslinking density of the blendings, although PI, which has high heat resistance, was included. Figure 5b,d illustrates the TGA and DTG analysis results of the postcured EPI. The postcured EPI samples then showed one mass decrease at about 345–364 °C. A postcuring process of 1 h or more induced the removal of the NMP residue of EPI and the reaction of the unreacted epoxy, which induced an increase in the crosslinking density, which is considered to improve heat resistance.

### 3.5. Mechanical Properties of EPI

The flexural and impact strengths of EPI are shown in Figure 6 and Figure 7, respectively. In addition, Equations (9) and (10) were used to calculate the flexural and impact strengths.
(9)$$S=\frac{3P_{max}L}{2bd^{2}}\;\mathrm{(Pa)}$$
(10)$$S=\frac{J}{A}\;\mathrm{(Pa)}$$
where *P_max_* (N) was the maximum load applied to the specimen, *L* (m) was the span distance of the UTM, *b* (m) was the length of the specimen, *d* (m) was the thickness of the specimen, *J* was the energy of the hammer used, and *A* (m) was the area of the specimen excluding the notch.

The results indicated that EPI’s flexural strengths increased with PI content: from 110.35 MPa for the neat EP to 171.77 MPa at the 50 phr condition. The values of the impact strength of EPI also exhibited a similar tendency. They increased continuously from 16.64 J/m for neat EP to 38.16 J/m for the EPI-5 sample. For the EPI-5 specimen, the flexural strength and the impact strength increased 55.66% and 129.33%, respectively, compared to those of the neat EP. The flexural strength of the postcured EPI increased with an increasing postcuring time (171.77–271.20 MPa). On the other hand, the impact strength of the postcured EPI showed a tendency to decrease (38.16–15.44 J/m). Compared to the EPI-5 specimen, the flexural strength of the EPI-5-10 specimen increased by 57.89% and the impact strength decreased by 59.54%.

The results of the flexural and impact strength of EPI can be inferred from previous results. As confirmed from the DSC results, the reaction between the epoxide groups of EP and the hydroxy groups of PI occurred rapidly. Then, it was crosslinked by reacting with amine groups of DDM (the hardener). As a result, the crosslinking density of the final blend decreased, which led to high ductility. High-stress resistance to bending fracture was observed in all blend samples compared to the pure EP. As the content of PI increased, the total energy to resist the deformation that each blend was able to absorb increased, which in turn improved the flexural and impact strength of the blend. On the other hand, the postcuring process increased the crosslinking density of EPI, leading to high stiffness. This improved the flexural strength, but the brittleness due to the high crosslinking density caused a decrease in the impact strength of the blend.

SEM images of the impact fracture surfaces are shown in Figure 8. The fracture surface of EP showed few wave patterns, whereas the fracture surface of EPI-5 showed several wave patterns. This suggests that the internal resistance of stress propagation in the blend progressed several times. On the other hand, the fracture surface of EPI-5-10 (high brittleness), which had a long postcuring time, did not resist impact, so it was confirmed that there were no patterns.

## 4. Conclusions

In this study, the effect of polyimide (PI) content change and postcuring time on the thermal and mechanical properties of EP were investigated. As the content of PI increased to 10–50 phr, the crosslinking density of EPI was lowered, resulting in ductility and an increased flexural and impact strength. On the other hand, as the postcuring time increased from 1–10 h, the crosslinking density of EPI increased, which improved heat resistance and stiffness, which increased the flexural strength but decreased the impact strength. In conclusion, EP and PI blending can improve mechanical strength, and the postcuring of EPI can improve heat resistance and flexural strength.

## Figures and Tables

**Figure 1 polymers-15-01072-f001:**
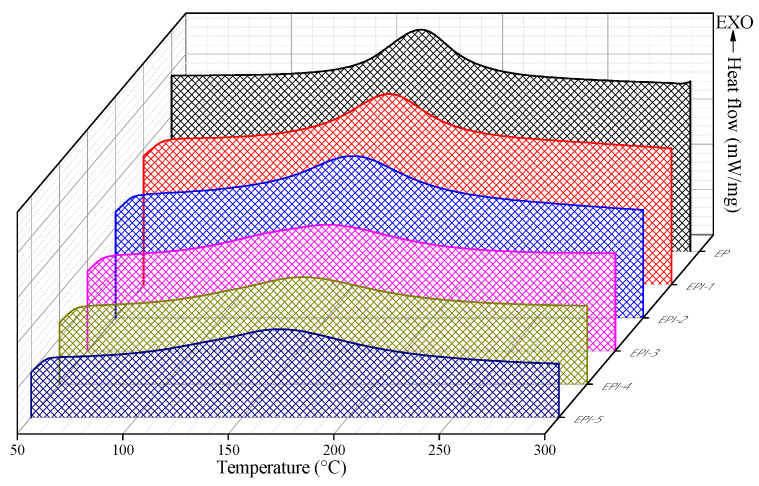
Curing temperatures of epoxy/polyimide blends analyzed as a function of polyimide content in N_2_ gas atmosphere.

**Figure 2 polymers-15-01072-f002:**
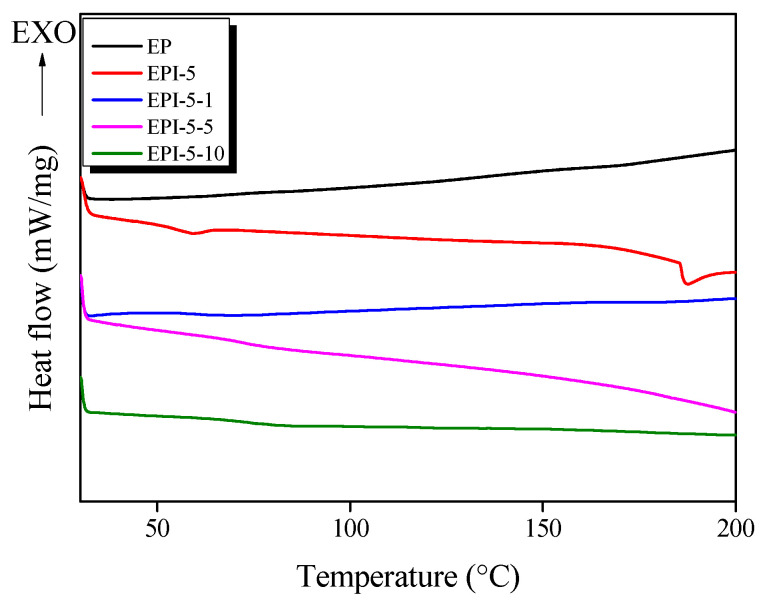
DSC thermograms of postcured epoxy/polyimide blends analyzed in N_2_ gas atmosphere.

**Figure 3 polymers-15-01072-f003:**
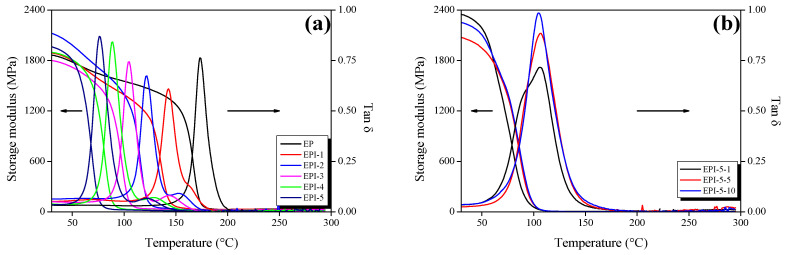
Storage moduli and tan δ values of cured epoxy/polyimide blends by dynamic mechanical analysis (DMA) in N_2_ gas atmosphere; (**a**) cured EPI and (**b**) postcured EPI.

**Figure 4 polymers-15-01072-f004:**
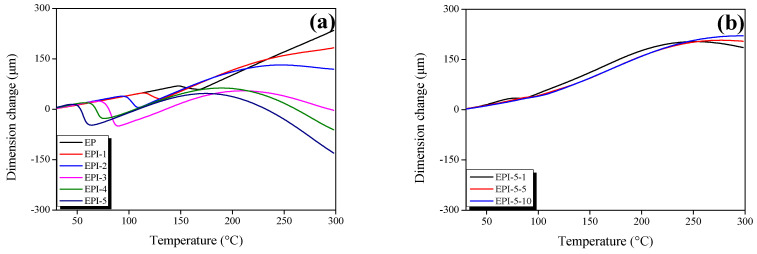
Thermomechanical analysis (TMA) thermograms of cured epoxy/polyimide blends analyzed in N_2_ gas atmosphere; (**a**) cured EPI and (**b**) postcured EPI.

**Figure 5 polymers-15-01072-f005:**
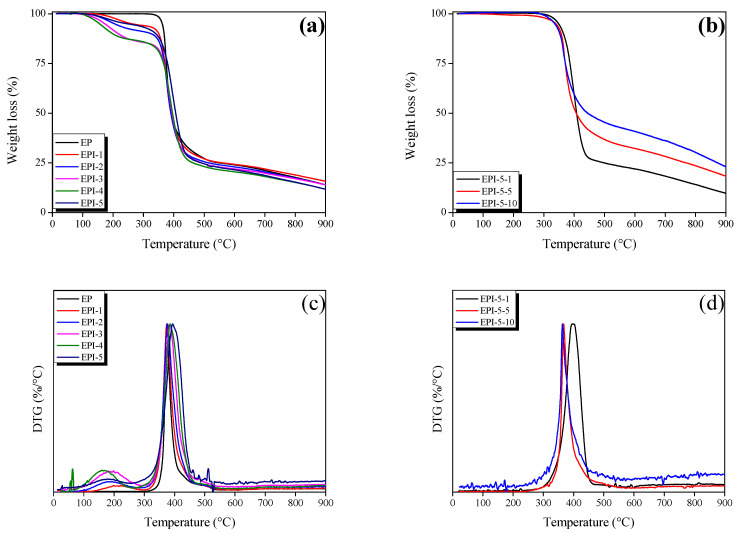
Thermogravimetric analysis (TGA) and DTG thermograms of cured epoxy/polyimide blends analyzed as a function of polyimide content in N_2_ gas atmosphere; (**a**,**c**) cured EPI and (**b**,**d**) postcured EPI.

**Figure 6 polymers-15-01072-f006:**
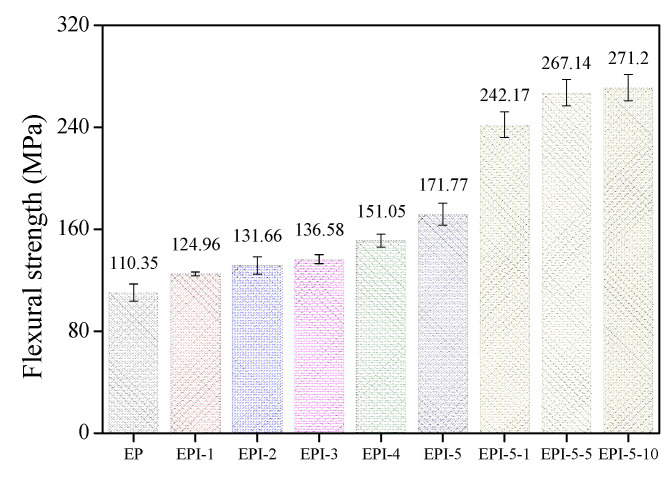
Flexural strengths of cured epoxy/polyimide blend composites.

**Figure 7 polymers-15-01072-f007:**
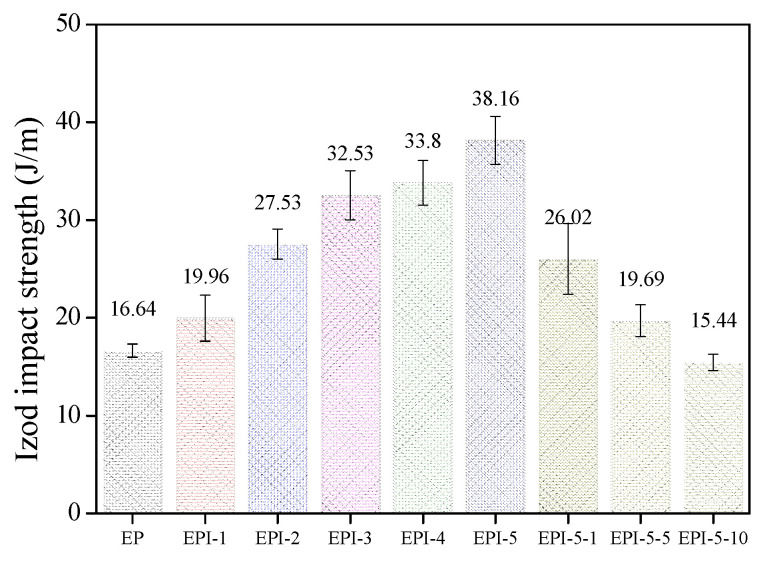
Izod impact strengths of cured epoxy/polyimide blend composites.

**Figure 8 polymers-15-01072-f008:**
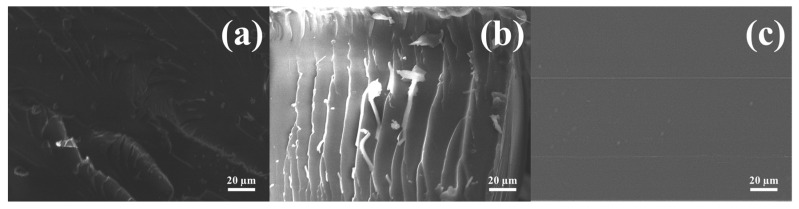
Scanning electron microscopy (SEM) images on the impact fractured surface of cured epoxy/polyimide composites: (**a**) EP, (**b**) EPI-5, and (**c**) EPI-5-10.

**Table 1 polymers-15-01072-t001:** DSC parameters of postcured epoxy/polyimide blends analyzed in N_2_ gas atmosphere.

Nomenclature	T_g_ (°C)	* T_s_ (°C)
EP	160.71	-
EPI-5	55.67	187.47
EPI-5-1	59.27	-
EPI-5-5	72.66	-
EPI-5-10	74.52	-

* T_s_: softening temperature.

**Table 2 polymers-15-01072-t002:** Crosslinking density of cured epoxy/polyimide blends by dynamic mechanical analysis (DMA) in N_2_ gas atmosphere.

Sample	Crosslinking Density (10^3^ mol/m^3^)	T_g_ (°C)
EP	2.46	173.47
EPI-1	2.40	142.77
EPI-2	2.05	121.47
EPI-3	1.71	104.52
EPI-4	1.33	88.56
EPI-5	1.18	76.32
EPI-5-1	2.45	105.85
EPI-5-5	2.67	106.30
EPI-5-10	3.76	104.49

**Table 3 polymers-15-01072-t003:** Thermal expansion coefficient of cured epoxy/polyimide blends by thermomechanical analysis (TMA) in N_2_ gas atmosphere.

Sample	Before T_g_ CTE (10^−6^ μm/μm·°C)	After T_g_ CTE (10^−6^ μm/μm·°C)	* T_s_ (°C)
EP	68.27	169.22	-
EPI-1	67.71	113.23	224.52
EPI-2	67.09	75.43	215.60
EPI-3	65.20	27.96	206.30
EPI-4	65.26	−18.90	202.23
EPI-5	72.29	−44.57	186.30
EPI-5-1	98.44	188.30	235.13
EPI-5-5	81.16	181.40	245.15
EPI-5-10	74.44	174.40	248.84

* T_s_: softening temperature.

**Table 4 polymers-15-01072-t004:** Thermal parameters of postcured epoxy/polyimide blends analyzed in N_2_ gas atmosphere.

Sample	^1^ IDT (°C)	^2^ T_max_ (°C)	^3^ A*·K*	^4^ IPDT (°C)
EP	361.89	374.54	0.6313	639.07
EPI-1	263.42	374.14	0.6627	667.39
EPI-2	203.03	376.78	0.6147	619.51
EPI-3	172.84	385.21	0.6090	614.46
EPI-4	155.23	385.33	0.5552	560.94
EPI-5	245.81	393.78	0.5785	584.13
EPI-5-1	348.46	398.63	0.6602	598.94
EPI-5-5	339.41	367.33	0.8517	769.53
EPI-5-10	336.12	363.32	0.9948	895.39

^1^ IDT: initial degradation temperature; ^2^ T_max_: temperature for maximum rate of decomposition; ^3^ A*·K*: thermal stability factor; ^4^ IPDT: integral procedural decomposition temperature.

## Data Availability

Not applicable.

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
