# Peer review of "Thermal and Mechanical Characterization of Epoxy/Polyimide Blends via Postcuring Process"

_polymers, 2023, doi:10.3390/polym15051072_

Round 1

Reviewer 1 Report

The article presents an interesting topic of research. The article presents the results of thermal and mechanical tests of the Epoxy/Polyimide mixture. The article could be published but after improvement.

The review of the literature does not fully correspond to the scope of the presented research.

Similar studies by other authors should be taken into account and the need to perform the presented studies should be indicated to a greater extent.

The manufacturing process of the tested materials should be described in detail, specifying the machines, devices and parameters.

The text needs to be carefully checked and corrected in accordance with the guidelines of the journal (for example, spacing before °C and the like), („Figure S3” ?).

In the formulas throughout the work, units are not given, also in the symbols,

designations of formulas should be exposed from the text

Figure 2, 1 - the scale (Heat flow) should be completed

Figure 3, 4, 8 – (a) and (b) are misplaced

Figure 8c is of poor quality

Characteristic parameters of materials are not calculated from DSC thermograms. The research results should be supplemented, for example, in tabular form and discussed in the text.

Why were dynamic properties not tested at different excitation frequencies (fig. 3b)?

Whether the specified Tg values are appropriate in this regard (table 1)?

Why the DTG was not presented, which would allow for a broader analysis?

Reviewer 2 Report

Manuscript Number: Polymers-2190324

The manuscript written by Lee et al. titled “Thermal and Mechanical Characterization of Epoxy/Polyimide Blends via Post-Curing Process” provided a study of the utilization of long-chain polyimide as a crosslinker to enhance the strength of the DGEBA resin cred networks. I want to suggest the following minor edits to make it suitable for Polymers.

1.      PI has been utilized to reduce the brittle nature of the DGEBPA-based resin, in the introduction, the author should write about such prior research, findings, and the significance of this research work.

2.      Previous extensive curing studies and network property evaluation of DGEBA resin had been carried out by Webster et al. using various curatives. The author could include such relevant research in the discussion. ( https://doi.org/10.1016/j.polymer.2021.124191, https://doi.org/10.1016/j.porgcoat.2020.105898)

3.      In the abstract the authors could include some numbers related to the key findings.

4.      Any information on the equivalent weight of the PI, please include.

5.      Line 122 “….it is physical properties are directly affected” should be “..it’s physical properties are directly affected”.

6.      Line 153 “please change the word ‘lowness’ to the appropriate one.

7.      Line 161 “It is judged that the cured EP, EPI-5, and post-cured EPI specimens were all completely cured, and no exothermic peak was observed.” I fit is true then why does the crosslink density of EPI-5 increase with post-curing?

8.      In figure 3 a, with increasing the PI from 1 to 4 there is a decrease in the Tan delta value, however, the crease in XLD should increase the loss modulus and thus the Tan delta value as seen in the case of EPI-5. Any explanation?

9.      If EPI-5 is fully cured then it is very unlikely that the PI would provide plasticization. Therefore the observed endotherm in the Tg of the EPI-5 cured network. I can also observe an endotherm for EPI-5 at around 60 °C, any explanation for this lower temperature endotherm? Also, table 1 shows a Tg for the EPI-5 around 76.32 °C.

10.  In table 1, it is strange that the crosslink density increases from 2.45 to 3.76 with post-curing from 1 to 10 hr, however, the Tg of the cured network decrease while it should increase. Any clarification?

11.  Please try to provide the best possible answers to the queries to make the paper more interesting.

Round 2

Reviewer 1 Report

The article may be published in its current form